# Investigating Potential Effects of Ultra-Short Laser-Textured Porous Poly-ε-Caprolactone Scaffolds on Bacterial Adhesion and Bone Cell Metabolism

**DOI:** 10.3390/polym14122382

**Published:** 2022-06-12

**Authors:** Emil Filipov, Liliya Angelova, Sanjana Vig, Maria Helena Fernandes, Gerard Moreau, Marie Lasgorceix, Ivan Buchvarov, Albena Daskalova

**Affiliations:** 1Institute of Electronics, Bulgarian Academy of Sciences, 72 Tzarigradsko Shousse Blvd., 1784 Sofia, Bulgaria; lily1986@abv.bg (L.A.); albdaskalova@gmail.com (A.D.); 2Faculdade de Medicina Dentaria, Universidade do Porto, Rua Dr. Manuel Pereira da Silva, 4200-393 Porto, Portugal; sanjana@fmd.up.pt (S.V.); mhfernandes@fmd.up.pt (M.H.F.); 3LAQV/REQUIMTE, University of Porto, 4160-007 Porto, Portugal; 4Laboratoire des Matériaux Céramiques et Procédés Associés, Université Polytechnique Hauts-de-France, INSA Hauts-de-France, CERAMATHS, F-59313 Valenciennes, France; gerard.moreau@uphf.fr (G.M.); marie.lasgorceix@uphf.fr (M.L.); 5Faculty of Physics, St. Kliment Ohridski University of Sofia, 5 James Bourchier Blvd., 1164 Sofia, Bulgaria; ivan.buchvarov@phys.uni-sofia.bg

**Keywords:** ultra-short laser processing, biomaterials, 3D printing, cell adhesion, antibacterial surfaces

## Abstract

Developing antimicrobial surfaces that combat implant-associated infections while promoting host cell response is a key strategy for improving current therapies for orthopaedic injuries. In this paper, we present the application of ultra-short laser irradiation for patterning the surface of a 3D biodegradable synthetic polymer in order to affect the adhesion and proliferation of bone cells and reject bacterial cells. The surfaces of 3D-printed polycaprolactone (PCL) scaffolds were processed with a femtosecond laser (λ = 800 nm; τ = 130 fs) for the production of patterns resembling microchannels or microprotrusions. MG63 osteoblastic cells, as well as *S. aureus* and *E. coli*, were cultured on fs-laser-treated samples. Their attachment, proliferation, and metabolic activity were monitored via colorimetric assays and scanning electron microscopy. The microchannels improved the wettability, stimulating the attachment, spreading, and proliferation of osteoblastic cells. The same topography induced cell-pattern orientation and promoted the expression of alkaline phosphatase in cells growing in an osteogenic medium. The microchannels exerted an inhibitory effect on *S. aureus* as after 48 h cells appeared shrunk and disrupted. In comparison, *E. coli* formed an abundant biofilm over both the laser-treated and control samples; however, the film was dense and adhesive on the control PCL but unattached over the microchannels.

## 1. Introduction

As one of the most common types of injury, bone fractures require novel approaches and constant improvements in order to build on the current tissue-repairing standards. Such standards include the use of metallic implants (titanium and its alloys, stainless steel, or cobalt chrome) that exhibit outstanding mechanical properties. However, there are still some disadvantages that limit the performance of these materials: (i) the stress-shielding effect that could lead to bone resorption; (ii) the release of ions that could be toxic to the host cells; (iii) the generally smooth surface of the implants representing a favorable site for bacterial adhesion; and (iv) the lack of biodegradability concerning non-load-bearing implants [1,2]. The need for novel materials to overcome these limitations has brought attention to the degradable biomaterials of either natural or synthetic origin. The chemistry of such materials (e.g., polymers) can be altered, which allows for the fine-tuning of their physical and mechanical properties [3]. An important aspect of the comparison between metals and polymers is that the latter can be enzymatically degraded to their building components and excreted or metabolized by the body in the process of tissue regeneration without the need for secondary surgery for removing the material [4].

Polycaprolactone (PCL) is a synthetic polymer with a wide application in tissue engineering. It is a semicrystalline polyester with a melting point between 59 °C and 64 °C [5]. Due to its low glass transition temperature of −56 °C, the polymer remains highly permeable for macromolecules under physiological conditions. Because of this property, PCL has been used as a material for drug-delivery systems [3]. Similar to PLA, the high number of methylene groups makes the polymer hydrophobic, which could affect its interaction with human cells [6]. PCL exhibits a very slow degradation rate that arises from its hydrophobicity and degree of crystallization. The molecular weight further influences these two properties of the material [7]. The degradation of PCL mainly occurs via hydrolysis either in an enzymatic (by esterases and lipases) or a non-enzymatic manner [8]. The process of polymer degradation yields intermediate products that could be metabolized via the citric acid cycle and thus eliminated from the body [5]. It has been investigated that PCL’s mechanical properties are not sufficient to withstand the tension that, for example, cortical bones could exert on it [9,10,11]. Thus, different strategies for the improvement of these properties have been employed, e.g., creating blends with ceramic materials or producing PCL scaffolds with distinct pore sizes and geometries [11].

The initial attachment of single bacterial cells to the implant’s surface is followed by the aggregation of microorganisms and the secretion of extracellular polymeric substances (EPS) [12]. The gradual formation of a thick slime layer represents the bacterial biofilm, which plays the role of a shelter for the bacterial cells. Thus, they can not only evade host immune cells but also resist antibiotic treatment that can lead to a major irreversible infection [13]. Bacteria can sense the surrounding environment via several mechanisms including chemical, biological, and physical [14]. The ionic concentration and the presence of biological molecules or antimicrobial peptides can guide the bacterial cells to the appropriate site for adhesion. Furthermore, the cells can use mechanosensors in order to make physical contact with different surfaces. The interactions between cells and surfaces fall into three main categories: non-specific physiochemical, specific, and surface mechanosensing [14]. On the material’s side, the factors that have a great influence on the bacterial adhesion include the chemical composition, surface energy, wettability, and surface topography [12,15]. The morphological appearance and the roughness of a material at the micro or nano level could physically hinder the attachment of a cell. Patterns with a distinct morphology (e.g., pillar-like) can lead to a reduced contact area and points for contact, thus preventing the cell from properly expanding its mechanosensors and establishing a physical connection with the material through specific molecules called adhesins [15]. Hence, various strategies for the modification of implants’ surfaces have emerged including the development of layers that either contain antimicrobial agents or aim to change the roughness of the implant.

A very prominent solution for developing a physical barrier for the inhibition of bacterial adhesion onto materials is femtosecond (fs)-laser surface processing [16]. Unlike the linear absorption of laser energy in metals, in the case of PCL, which is considered a dielectric, the laser energy is absorbed via non-linear processes, which involve multiphoton ionization [17]. Since the duration of an ultra-short pulse is less than the time (<ps) required for the pulse energy to dissipate along the structural lattice after a multiphoton ionization, the irradiation does not lead to a thermal diffusion that could, in turn, result in unwanted photomechanical damage to the material [18]. Thus, fs-laser micromachining allows for a gentle surface patterning that alters the roughness and the morphological appearance of the irradiated zone without substantially affecting its structural integrity [16]. Several research groups report the use of an fs-laser for surface texturing of materials [19,20,21]. Chen et al. investigated the effects of surface topography by creating parallel microchannels (width of 1–2 μm) lined with nanopillars (200 nm) (λ = 1040 nm; τ = 375 fs; ν = 20 kHz; E = 80 nJ) on borosilicate glass [20]. The study reported that the nanoroughness greatly reduced the viability of both *E. coli* and *S. aureus* by exerting mechanical stress on their membranes. Jalil et al. studied how the increase in laser fluence (0.1 J/cm^2^ to 3 J/cm^2^) with a shorter pulse width (30 fs) led to the gradual transition of ordered (laser-induced periodic surface structures; LIPSS) to disordered structures at both micro- and nanoscales on gold [21]. The authors observed that laser-induced nano topography (LIPSS) inhibited bacterial adhesion by reducing the contact points and disrupting the bacterial cells. In addition to their ability to hinder bacterial attachment, such types of superficial modifications could be used to influence the physiochemical properties of materials such as the wettability degree without greatly altering their chemical composition. For example, a study employing fs-laser parameters similar to the ones described in the following sections (λ = 800; τ= 130 fs; ν = 1 kHz) showed that the patterning of micro trenches with roughness on a nanoscale led to a significant increase in hydrophobicity (θ = 120°~156°) compared to unprocessed surfaces (θ = 75°) [22]. This process is mainly attributed to the formed topographies that influence the contact between liquids and the modified surface [23].

In this paper, we aim at developing antibacterial surfaces for bone tissue engineering. The novelty behind our approach for achieving this aim includes the use of ultra-short laser pulses for processing biocompatible polymers, which represents a contactless, highly precise, and reproducible method for obtaining surface patterns with defined locations and dimensions. The methodology described in this article involved the production of 3D-printed PCL scaffolds with pre-defined geometry, whose surface was textured via fs-laser modification. The antimicrobial potential of the formed patterns was evaluated by using *S. aureus* and *E. coli*. The effects of the modified scaffolds on the viability, proliferation, and differentiation of osteoblastic cells were further investigated.

## 2. Materials and Methods

### 2.1. Fabrication of Polymeric Scaffolds

PCL-based 3D scaffolds, resembling a woodpile construct, were fabricated via extrusion 3D printing. The standard operating procedure was already described and established by Daskalova et al. [24]. In brief, PCL pellets with M_n_ = 45 kDa (Sigma-Aldrich, St. Louis, MO, USA) were melted at 70 °C in a cartridge unit and extruded through a 250 μm needle at a pressure of 5 bar and speed of deposition of 95 mm/min. The constructs were fabricated layer by layer with space between separate fibers of 140 μm and a thickness of the layers of 160 μm. The layer deposition angle was set to 90 °C.

### 2.2. Surface Modification by Femtosecond-Laser Machining

The surface of the PCL scaffolds was processed with a Ti:sapphire fs-laser (Quantronix-Integra-C, Hamden, CT, USA) emitting at a central wavelength of 800 nm with a pulse duration (τ) of 130 fs. The repetition rate for (ν) was set at 1 kHz. The number of applied laser pulses (N) was adjusted by the velocity of a vertical translation stage controlled by computer software (LabView). All scaffolds were adjusted on the translation stage (moving in XY directions), which was positioned perpendicular to the direction of the laser beam. The parameters chosen for material processing for all in vitro experiments were as follows: N = 2 or 10; laser fluence (F) 0.08 J/cm^2^ and distance between two adjacent laser spots (d_x_) 45 or 35 μm. This decision was based on the findings of a previous investigation into this subject [24]. In addition to these samples, a separate set of PCL scaffolds were modified with the following fs-laser parameters: N = 1/2/5/10; F = 0.08/0.17/0.42/1.23 J/cm^2^; d_x_= 65 or 75 μm. These samples were used to assess the morphological changes that a gradual increase in the number of applied laser pulses and laser fluence could induce in the material.

### 2.3. Characterization of fs-Laser-Processed Scaffolds

#### 2.3.1. Assessment of fs Laser Derived Topography Features and Their Effects on the Morphology and Cell Growth Pattern of *S. aureus* and MG-63 Osteoblastic Cell Lines via Microscopic Techniques

Laser-processed matrices were visualized by scanning electron microscopy (SEM) (SU5000 Hitachi High-Tech Europe). Prior to the analyses, the samples were coated with a thin layer of platinum (~4 nm). The images were obtained at 15 kV. The morphological changes on the surface of the materials arising from the increase in applied laser pulses were investigated by a 3D optical surface metrological system Leica DCM3D (Leica Microsystems, Wetzlar, Germany). Two- and three-dimensional topographical images in true colors were obtained by vertical scanning of selected modified or control areas. All images were obtained at 20× magnification with the field of view being 636.61 × 477.25 μm^2^ or at 10× (field of view 1.27 × 0.95 mm^2^). Based on the acquired images in true colors, the roughness of the modified areas on the material was evaluated in accordance with ISO 4287. For this purpose, the arithmetical mean height (Ra) was taken into account. All data obtained by the 3D optical system was processed via ProfilmOnline software (www.profilmonline.com, accessed on 6 April 2022).

All samples used for in vitro analyses with osteoblast-like and bacterial cells were fixed in 1.5% glutaraldehyde solution (prepared in 25% cacodylate solution, TAAB laboratories equipment Ltd., Aldermaston, England) for 15 min followed by storage in sodium cacodylate solution. They were serially dehydrated in gradually increasing alcohol concentrations (50%, 70%, 90%, 100% ethanol) followed by critical point drying. Samples were sputter-coated with a thin layer of gold-palladium coating and evaluated using SEM (JEOL JSM-700, Tokyo, Japan).

#### 2.3.2. Analysis of Wettability and Surface Energy Changes in Relation to fs-Laser Processing

Wettability studies were performed using a video-based optical contact angle measurement device DSA100 Drop Shape Analyzer (KRÜSS GmbH, Hamburg, Germany). In order to fully evaluate the wetting state and the total surface energy of the control and modified surfaces, three types of liquids were used based on their polarity: distilled water (highly polar), ethylene glycol (medium polarity), and diiodomethane (very low polarity). Static contact angles were measured at room temperature by the sessile drop method on droplets of 2 μL. The measurements were performed with a minimum of 3 drops per sample type. The drop evolution was followed for a total of 3 min as measurements were taken at each second during the first minute and every minute for the next two. Contact angles and surface energy were calculated by ADVANCE software (KRÜSS GmbH, Hamburg, Germany) fitting the drop profiles to the Young–Laplace equation. Surface energy (SFE) was calculated by the software following the Owens–Wendt–Rabel–Kaelble (OWRK) equation (Equation (1)) [25].
(1)$$\gamma_{12}=\gamma_{1}+\gamma_{2}-2\sqrt{(\gamma_{1}^{d}}\gamma_{2}^{d})-2\sqrt{(\gamma_{1}^{p}}\gamma_{2}^{p}),$$

### 2.4. Degradation Test in Phosphate Buffer Saline

Fs-laser-irradiated (F = 0.08 J/cm^2^; N = 2) 3D printed PCL scaffold was immersed in phosphate buffer saline (PBS, pH 7.2, Sigma-Aldrich, St. Louis, MO, USA) for 7 weeks. The pH was read every week with a pH meter with an external sensor (VAT1011, V & A Instrument Co., Ltd., Shanghai, China).

### 2.5. In Vitro Cytocompatibility Assessment

In vitro cytocompatibility assessment was performed on osteoblast-like MG63 (ATCC^®^CRL-1427™) cells cultured in alpha-MEM medium supplemented with 10% fetal bovine serum (FBS), 100 IU/mL penicillin, 100 µg/mL streptomycin, and 2.5 µg/mL amphotericin B (all reagents from Gibco, USA) at 37 °C, 95% humidity and 5% CO_2_ atmosphere. Cells were seeded over the materials at a density of 2 × 10^5^ cells/cm^2^ and cultured for 7 days in basal (as described above) and osteogenic-induced conditions. On induced cultures, MG63 cells were pre-treated with 50 μg/mL ascorbic acid and 10 nM dexamethasone for 48 h. Next, cells were passaged and seeded over the materials in an osteogenic medium containing 50 µg/mL ascorbic acid, 10 nM dexamethasone, and 10 mM beta-glycerophosphate (all from Sigma-Aldrich, St. Louis, MO, USA). Cell cultures were characterized for metabolic activity (Resazurin assay), alkaline phosphatase (ALP) activity, and SEM analysis.

#### 2.5.1. Resazurin Assay

Metabolic activity of MG63 cells on PCL samples was assessed through the Resazurin assay on day 1, day 3, and day 7 for both non-induced and induced cell cultures. Briefly, all samples were transferred to a fresh well plate before incubation for 3 h in 10% Resazurin solution (Resazurin sodium salt, Sigma-Aldrich R7017) prepared in a complete medium (alpha-MEM with 10% FBS, 100 IU/mL penicillin, 100 µg/mL streptomycin, and 2.5 µg/mL amphotericin B) at 37 °C. Fluorescence (530 nm excitation/590 nm emission) was measured in a microplate reader (Synergy HT, Biotek, Winooski, VT, USA) with Gen5 1.09 Data Analysis Software.

#### 2.5.2. Alkaline Phosphatase (ALP) Activity

The ALP activity of cells was evaluated on day 3 and day 7 of osteoinduction. Cell lysates were prepared in 0.1% Triton X-100 (in distilled water) for 30 min followed by the hydrolysis of p-nitrophenyl phosphate substrate (p-NPP, 25 mM, Sigma-Aldrich, USA) in an alkaline buffer (pH 10.3, 37 °C, 1 h). The reaction was stopped with 5 M NaOH and the product (p-nitrophenol) was measured at 400 nm in a microplate reader (Synergy HT, Biotek, USA). Results were normalized to total protein content, measured using the DCTM Protein Assay (BioRad, Hercules, CA, USA) according to the manufacturer’s instructions, and expressed as nanomoles of p-nitrophenol per microgram of protein (nmol/µg protein).

### 2.6. Antimicrobial Activity

Antibacterial activity of all samples was assessed against *Staphylococcus aureus* (ATCC 25923) and *Escherichia coli* (ATCC 25922). PCL samples were incubated with 10,000 CFU/mL of bacterial suspension for 6 h, 24 h, and 48 h in tryptic soy buffer. Biofilm formation/inhibition was assessed using SEM analysis. CFU assay measurements were made after 24 h of incubation.

### 2.7. Statistical Analysis

Results for the biological assays are presented as mean ± standard deviation of three independent experiments, with three replicas for each experiment. One-way analysis of variance (ANOVA) was used with Bonferroni’s post hoc test for data evaluation. Values of *p* ≤ 0.05 were considered significant.

## 3. Results

### 3.1. Inducing Morphological Changes in the Surface Profile of the 3D-Printed PCL Scaffolds by fs-Laser Micromachining

The physical effects that the fs-laser treatment exerted on the surface of the scaffolds were visualized by a 3D optical system and SEM. As previously seen, the chosen working parameters F = 0.08 J/cm^2^ and N = 2 and 10 lead to the formation of surface patterns resembling protrusions (N = 2) and microchannels (N = 10) (Figure 1) [26]. The average height of the protrusions was 12.03 μm, whereas the average depth of the microchannels was 32.87 μm with an average width of 28.7 μm. The surface roughness within the microchannels ranged between 0.14 and 0.8 μm. A measurement of the arithmetical mean height (Ra) across the modified areas was obtained in order to observe the variation of the roughness in relation to the achieved surface morphologies. For N = 2, the Ra value was 1.15 μm and for N = 10 it was found to be 5.1 μm. In comparison, the control sample showed an Ra of 0.08 μm. A study monitoring the attachment and proliferation of MG63 cells on Mg-based metallic glass showed that a surface with a roughness of ~0.22 μm led to the highest cell attachment, whereas a roughness of ~0.24 μm led to an improved calcium deposition [27]. The formed microchannels as a result of the increased number of applied laser pulses, substantially improved the overall roughness of the material, thus providing a surface that would be suitable for cell adhesion and osteogenic differentiation.

As an additional assessment, the evolution of the ablation crater on the material in relation to the number of applied laser pulses and laser fluence was investigated. Regardless of the laser fluence used in the experiments, with a lower number of applied pulses (1 or 2) the laser–matter interaction led to a build-up of melted material without any substantial ablation (Figure 2a,b). However, the increase of N to 5 and 10 caused the formation of engraved channels (Figure 2c,d). The morphological observations of the 3D true-color images revealed the transition between the protruding patterns and the microchannels.

As expected, the highest laser fluence (1.25 J/cm^2^) formed a laser spot with the largest diameter with both N = 1 and N = 10 (Figure 3). However, the increase in the fluency in the range 0.08–0.42 J/cm^2^ at N = 1 did not lead to a major change in the dimensions of the physical mark on the material.

### 3.2. Hydrophobic Behaviour of fs-Laser-Treated PCL Scaffolds

The water contact angle (WCA) evaluation with a highly polar solvent revealed that the fs-laser treatment with N = 10 led to an increase in the wettability of the polymeric scaffolds; however, the material still retained a contact angle of slightly less than 90° 180 s after the drop deployment (Figure 4 and Figure 5). In contrast, the control samples showed a WCA of 123.3° at t = 1 s and 115.8° at t = 180 s. When comparing the two types of formed surface morphologies, it could be clearly seen that the microchannels had a stronger effect on the improvement in PCL wettability. The reason behind this observation could be that a part of the water droplet spread along the microchannel while entrapping air in the valleys of the channels. Thus, a solid–liquid–air interface could be formed that would not allow the full disruption of the droplet. In this case, the interaction between the liquid and the roughened surface might follow the Cassie–Baxter wetting state [28]. Potentially, the microprotrusions had the same impact on the water droplet, in terms of spreading, as they resembled continuous channels due to their partial fusion. By observing the laser-induced patterns and the wettability results, we could hypothesize that the reduction in the water contact angle values was attributed mainly to the changed surface morphology.

Contrary to the results with distilled water, ethylene glycol drops fully spread over the surface of both the laser-treated and the control samples approximately 30 s after deposition (Figure 6). In the case of the highly dispersive solution (diiodomethane), the drops of the solvent immediately wetted the whole surface of both the modified and control scaffolds. Hence, a contact angle could not be measured on either of the two types of samples. Both ethylene glycol and diiodomethane possess a higher dispersive component of their surface energy compared to the polar one, and the same statement has been reported for polycaprolactone, mostly attributed to its nonpolar CH_2_ groups [29,30].The complete spread of both solvents on the scaffold’s surfaces could be explained by the attraction between the higher dispersive components of surface energies of both the material and the solvents [30]. The authors implied that a decrease in the polarity of a given surface, leaving a higher nonpolar component, would have a repelling effect against a polar solvent such as distilled water but not toward nonpolar solvents.

Based on the contact angles of distilled water and ethylene glycol, the surface energy of the scaffolds was calculated by the software. Overall, it was noted that the fs-laser irradiation greatly reduced the surface free energy of the material. The analysis of the control samples showed an SFE of 193.8 mN/m. On the contrary, the samples treated with N = 10 exhibited an SFE of 20.8 mN/n and the samples patterned with microprotrusions, 18.3 mN/m. These results were in contradiction to the ones regarding the wettability of the scaffolds. In particular, the higher surface energy would imply that a given surface is hydrophilic and a reduction in the surface energy would result in a shift to a hydrophobic state [30]. However, our findings showed the opposite trend—the laser-treated samples with lower values for SFE were hydrophilic (θ = 82.3° for the microchannels pattern and θ = 88.9° for the protrusions pattern), whereas the control scaffolds with a nearly 10-fold higher SFE were found to be hydrophobic (θ = 115.8°). Furthermore, the SFE was calculated by built-in software taking into consideration the contact angles of water and ethylene glycol. Unlike water, ethylene glycol completely spread over both the treated and control surfaces 30 s after drop deployment, which could have impacted the overall software calculation. Based on these observations and together with the presence of surface laser-induced roughness, it could be concluded that the method used for obtaining SFE values might induce an error in the overall calculation of SFE. Thus, further optimization of the experimental procedure is needed, and using alternative methods for the estimation of surface energy shall be applied.

### 3.3. Change in pH of PBS in Accordance with PCL Degradation

As a preliminary degradation test, the fs-laser-processed PCL was immersed in PBS over 7 weeks and the change in the pH of the solution was monitored. There was an overall reduction trend from a pH of 7.3 at week 0 to a pH of 6.72 at week 7 (Figure 7). The sharp peak at week 3 indicated that a slight increase in the pH values was not considered significant. Up to this point, no morphological analyses during the degradation test have been performed; however, no significant changes are expected. Due to its slow degradation rate (potentially taking up to 4 years), it has been shown that it would take about 12 weeks of PCL in PBS to exhibit initial slight disruptions in the surface morphology [31,32]. Thus, it has been assumed that our preliminary degradation test should not have induced any alterations in the laser-induced surface modifications. Further optimization of the experiment as well as performing it by using simulated body fluid would yield more accurate results.

### 3.4. Cytocompatibility

In order to assess the cytocompatibility of human osteoblast-like cells on fs-laser-induced microtopographies on PCL, cells were cultured for 11 days on the microchannel-patterned surface along with the non-laser treated controls (Figure 8). The laser-treated samples demonstrated significantly higher cell viability at all points during cell culture compared to the control samples. Further, SEM analysis revealed that cells showed a typical elongated morphology with cytoplasmic extensions and both cell–cell contact and cell–material interaction. Contrastingly, only a few cells with a round and disrupted morphology were seen on the non-laser treated controls. The distinct cell growth pattern and morphological behavior of the laser-treated and untreated samples are clearly seen in Figure 8e, which shows a representative image of a seeded semi-treated PCL sample.

Next, osteoblast induction with osteogenic growth factors (ascorbic acid, dexamethasone, and beta-glycerophosphate) was selectively performed on the two laser-induced topographies— the microchannels and protrusions (Figure 9). The resazurin assay revealed that cells demonstrated higher metabolic activity on protrusions compared to microchannels (statistically significant at day 3, ~20%, *p* ≤ 0.05). The same modification also led to a higher production of total protein content by the cells in comparison to the microchannels (not shown). However, the ALP activity (ALP levels normalized to total protein content) was significantly higher in microchannels (at day 7, ~twice, *p* ≤ 0.05). This suggests enhanced osteogenic differentiation of cells growing on the microchannels. Nevertheless, SEM analysis revealed cell adhesion and alignment along both microchannels and protrusions (Figure 9). Furthermore, induction of the cells with osteogenic factors led to the deposition of nanovesicles and extracellular matrix proteins that are visible at higher magnification, indicating the osteogenic potential of the cells in both structures.

### 3.5. Antibacterial Assay

*S. aureus* adhesion was observed on all PCL surfaces (microchannels, protrusions, and the non-laser treated control), but colonization appeared relatively limited. Key factors that could influence bacterial colonization include the physicochemical properties of the material’s surface as well as its roughness. Taylor et al. demonstrated that although an increase in surface roughness from 0.04 μm to 1.24 μm stimulated bacterial adhesion, a further rise in roughness beyond 1.86 μm resulted in reduced bacterial retention [33]. Our results are in accordance with these reports and show greater bacterial colonization on protrusions, whereas the surface roughness is below the threshold (Ra = 1.15 μm) compared to the microchannels (Ra = 5 μm) with a higher surface roughness than the threshold. On the control PCL samples, the SEM images suggested a slight increase in colonization from 6 h to 48 h (Figure 10a,b) with the cells maintaining the typical round morphology (Figure 10c) and size of ~ 0.5–1.0 μm in diameter. In contrast, bacterial cell behavior was negatively affected by the PCL laser-treated samples. At 48hrs after incubation, the number of attached cells on the treated samples was clearly lower (Figure 10e,h) than that on the control (Figure 10b). Laser-treated surfaces induced a significant disruption in the cell morphology, which was more evident in the microchannel topography (Figure 10f) compared to the protrusion surface (Figure 10i). Cells lost the typical round morphology, presenting a rough appearance and evident signs of cell lysis (Figure 10f,i). CFU analysis further revealed that the microchannels showed potential antibacterial activity compared to protrusions. The mean size of the surface topographical features has been correlated with the bacterial adhesion and colonization [34]. It has been established that when the size of the micron-scale topographies is lower than the size of bacteria, they prevent bacterial adhesion by limiting the surface area and contact points for the attachment [35]. In our study, the overall surface roughness within the channels was found between 0.14 and 0.8 μm, which is lower than the dimensions of the bacteria types tested in our study, *S. aureus* (0.5–1 μm) and *E. coli* (1–1.5 μm). This explains the greater inhibition of bacterial attachment on microchannels.

Considering the current visual and analytical results on surface morphology, a reason for this observation could be that the bottom of the microchannels (Figure 10e,f) appeared to have lower roughness due to the laser-induced material redistribution resulting in the partial movement of the surface layers and the subsequent cooling of the material. Thus, there would be less surface area for the bacterial cells to attach to. In comparison, the control sample and the one bearing microprotrusions appeared to have a rougher and more-caressed surface morphology, which could provide for more cells to adhere to. Furthermore, another factor that could influence the adhesion of bacteria is their motility. *S. aureus* is known to be a non-motile bacterium and a lack of motility could affect its successful spread and adhesion to the bottom part of the microchannels. A similar low adhesion of non-motile *P. fluorescence* has been observed by Scheuerman et al. who monitored the attachment of bacteria to microscale grooves on silicon [36].

The behavior of *E. coli* on the PCL samples was clearly different. SEM analysis revealed a significantly higher bacterial coverage compared to that of *S. aureus*. This resulted from the very high growth rate from 6 h to 48 h of incubation. At 6 h, adhesion was higher on the PCL control (Figure 11a) compared to that on the laser-treated surfaces (Figure 11e,i). In these topographies, cell adhesion was mainly observed in the protected bottom of the surface channels (Figure 11e) and on the smooth surface around the protrusions (Figure 11i). The bacteria presented with characteristic rod-like morphology and regular size (~1–2 μm long). During the incubation period, the high cell growth was associated with the formation of an extracellular matrix on all surfaces. Nevertheless, evident differences were seen in the formed biofilm on the untreated (control) and treated PCL surfaces. On the control PCL, the cells were already deeply embedded within a dense matrix at 24 h (Figure 11c) and partially buried at 48 h (Figure 11d). In contrast, the cells growing on the lasered surfaces formed a loose fiber reciprocal connection and were seen as individualized, being unable to produce a continuous sticky matrix (Figure 11g,h,k,l). Furthermore, anomalies in the cell division pattern were noted with the formation of long bacterial chains (Figure 11l).

## 4. Discussion

Cellular attachment and behavior are influenced by both the surface topography as well as the wettability of the scaffolds. Polycaprolactone (PCL) is an FDA-approved, hydrophobic polymer, which has been reported to have poor cell attachment. Our study demonstrated that femtosecond-laser-derived microchannels on the surface of PCL rendered the scaffold more hydrophilic by reducing the water contact angle from 115.8° to 82.3°, thus enhancing cellular adhesion and proliferation. These results are in accordance with previous studies reporting the enhanced laser-induced wettability of PCL and its correlation with cellular adhesion [24,37]. As an addition to its surface hydrophilicity, another important aspect of the behaviour of the PCL scaffold when in contact with the external environment is its stability over time and more specifically the endurance of the surface modifications. Due to the polymeric nature of the PCL, the resulting fs-surface modifications on the 3D constructs would remain solid and permanent as they do not change their morphology over time when stored at room temperature. The colonization of the surface with cells participating in the tissue repair process would take on average between 5 and 15 days, and it has been observed that substantial changes in the surface morphology of PCL in phosphate buffer saline (PBS) occur after 72 weeks. Thus, it could be stated that the laser-induced surface modifications would remain stable long after the initiation of tissue regeneration [32,38].

We generated femtosecond-laser-induced microstructures on PCL such as microchannels and microprotrusions, which promoted the directional and guided growth of osteoblast-like cells. We observed a higher degree of cell attachment and alignment along the microchannels with the enhanced osteogenic potential of cells (ALP activity). We further observed signs of differentiation including extracellular secretions on the microstructured PCL scaffolds indicating the osteogenic potential of the cells over the PCL scaffolds. These results are supported by past studies reporting enhanced attachment and guided cell growth along microchannels for various cell types [39,40,41]. A possible mechanism of microstructure-induced cell growth is postulated to be the increased surface roughness that supports protein adsorption and thus cellular adhesion [42]. Our analyses showed that the Ra values for the bottom of the laser-derived microchannels ranged between 0.14 and 0.8 μm. Andrukhov et al. explored the effects of microroughness on the osteogenic potential of MG63 cells [43]. This study indicated that titanium surfaces with an Ra of 1 μm enhanced the expression of ALP and osteocalcin, whereas Ra values of more than 2 μm reduced both the cell adhesion and differentiation. Our observations were also in accordance with the findings of Faia–Torres et al., whose study demonstrated that PCL membranes with an Ra of ~0.93 μm improved the levels of ALP expressed by human mesenchymal stem cells [44]. In the future, validation studies employing the proposed experimental methodology can be performed on human mesenchymal stem cells with a detailed analysis of gene expression.

Several cellular studies in the past have been performed on femtosecond-laser-induced microstructures on biodegradable polymers indicating guided cell growth and even differentiation into different lineages in the absence of inducing factors [40,45]. Yeong et al. demonstrated a high degree of alignment and proliferation for C2C12 mouse myoblast cells along the microchannels generated by femtosecond-laser on a poly(L-lactide-co-epsilon-caprolactone) copolymer [39]. Further studies on this co-polymer with human mesenchymal stem cells revealed an upregulation of the expression of myogenic genes [46]. Studies on other biopolymers such as collagen, gelatine, and elastin also demonstrated preferential cell proliferation and migration along laser-irradiated micropatterns [47]. Femtosecond-laser ablation has also been used as an approach for confining the growth of cells. Studies with mouse embryonic cell cultures revealed that fs-laser-ablated PCL-gelatine blends led to the confined growth of cells in the microwells [48]. This strategy has also been exploited for vascular tissue engineering for containing the growth of smooth muscle and endothelial cells along tubular PCL scaffolds [49].

In addition to enhanced cytocompatibility, the results also pointed to the antibacterial potential of the femtosecond-laser PCL surfaces. It is well-established that the adhesion, spreading, and growth of both eukaryotic and prokaryotic cells are greatly affected by surface topography. However, the responses to how they sense topographical features are distinct. Eukaryotic cells have the ability to expand the cytoplasm by adapting their morphology to the underlying surface thus retaining their shape, compared to bacteria that have a characteristic shape and a limited capacity to deform. This hinders bacteria–surface interaction and particular topographical features may further limit bacterial sensing ability, preventing bacterial adherence [50]. Antibacterial activity associated with topographical features is advantageous for its long-term effects as it retains its anti-adhesive effects and/or elicits mechano-inhibitory effects after bacteria attachment, i.e., morphological abnormalities and cell lysis [51]. Femtosecond-laser-induced microtopographies have been reported to influence bacterial cell attachment, colonization, and biofilm formation for both Gram-positive and Gram-negative bacteria [52,53,54]. Furthermore, the type of laser-induced pattern (such as pillar-like, honeycomb, etc.) has also been demonstrated to influence reduced bacterial attachment and confined entrapment [55,56,57]. In agreement with this, the present results on PCL surfaces clearly suggest that the femtosecond-laser-induced topographies, in particular the microchannel’s patterned surface, had a negative effect on the behavior of *S. aureus*, the most common Gram-positive bacteria associated with biofilm formation in orthopaedic infections. The behavior of *E. coli*, which was occasionally involved in these infections, was also clearly affected as the bacteriostatic effect of the scaffolds was less profound on *E. coli* and they showed reduced sensitivity to microtopography. On the laser-treated surfaces, sessile (adhered) bacteria lost the ability to form a sticky and dense polymeric matrix preventing the establishment of a stable biofilm, which is the most relevant 3D structure for bacteria to evade the host defence mechanisms [58]. Nevertheless, within the observed inhibitory effects, significant differences were noted in the behavior of the two species on the PCL surfaces, namely a high proliferation growth rate of *E. coli* compared to that of *S. aureus*. Previous studies also reported that bacterial responses to underlying surface topography are highly species- and strain-dependent, including in femtosecond-laser-modified surfaces [52,54]. These differences in the antibacterial activity of *S. aureus* and *E. coli* are related to the structure and composition of the cell wall in Gram-positive and negative bacteria. *S. aureus* has a thick, rigid peptidoglycan layer. In contrast, *E. coli*, which is a Gram-negative bacterium, has a thin peptidoglycan layer with an additional outer membrane offering extra resistance to the bacteria. In contrast to the bacteria, the cell membrane of mammalian cells is highly fluidic and can adapt easily to micron-scale topographies allowing cytoplasmic spreading and migration [59]. Cells attach to the substrate using integrin proteins and migrate and then as probing along a surface using filopodia and lamellipodia, which extend to the micron scale. Increased surface roughness provides greater focal adhesion points leading to higher migration and proliferation. This explains the differential responses of bacterial and mammalian cells (including osteoblastic cells) toward surface topography. Overall, the contrasting responses of the bacteria to the laser-processed surfaces are determined by various cell-specific factors. Some of these might include the intrinsic structural, biochemical, and metabolic differences between Gram-positive (*S. aureus*) and Gram-negative (*E. coli*) bacteria that, together with the distinct size and morphology of the two species, determine the specific bacteria/surface interactions during adhesion and subsequent proliferation.

## 5. Conclusions

In this study, we demonstrated a novel approach to patterning the surface of 3D polycaprolactone scaffolds by femtosecond-laser with the aim of developing distinct types of topographies—microchannels and microprotrusions. The parallel microchannels allowed the successful guidance and enhancement of the osteogenic potential of MG63 cells. In combination with the improved cytocompatibility, the same microtopography showed strong antibacterial effects against *S. aureus*. By developing a biodegradable scaffold that has the potential to simultaneously promote bone tissue regeneration while preventing bacterial biofilm formation, we become a step closer to overcoming the current problems in bone tissue engineering.

## Figures and Tables

**Figure 1 polymers-14-02382-f001:**
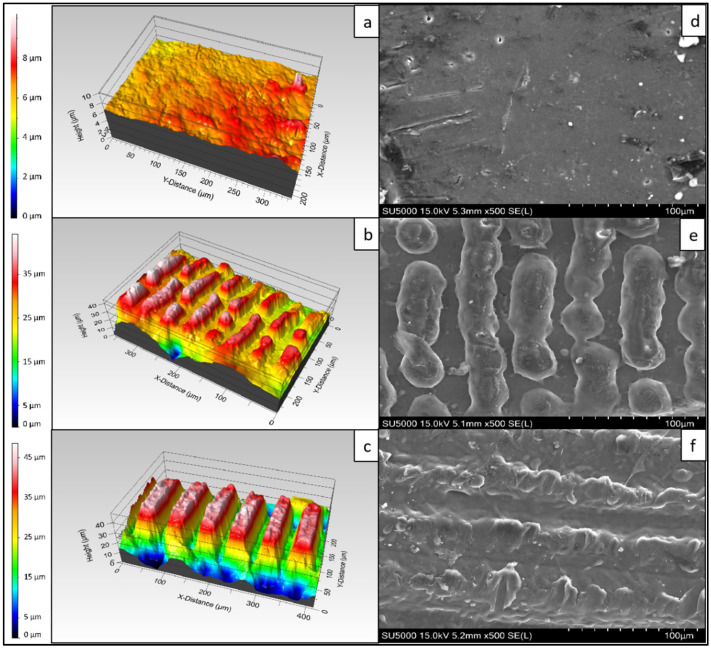
3D and SEM profiles of PCL scaffolds irradiated with an fs-laser. (**a**,**d**) Control sample; (**b**,**e**) A fiber of the polymeric scaffold processed with F = 0.08 J/cm^2^; N = 2; (**c**,**f**) A fiber of the polymeric scaffold processed with F = 0.08 J/cm^2^; N = 10.

**Figure 2 polymers-14-02382-f002:**
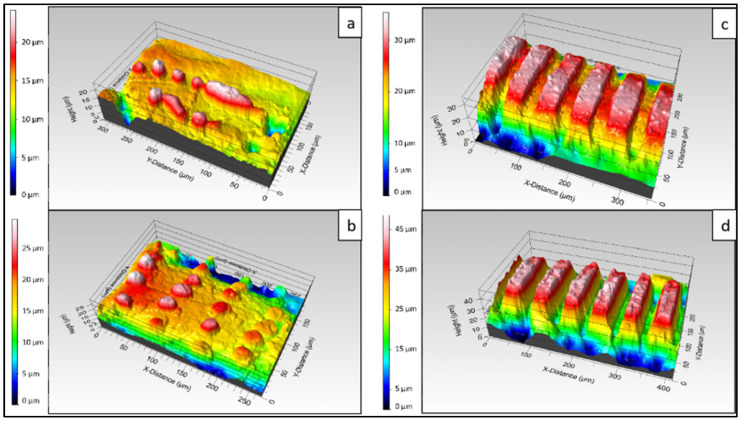
3D profiles of PCL scaffold treated with the same fluence (0.08 J/cm^2^) and rising applied laser pulses. (**a**) N = 1; (**b**) N = 2; (**c**) N = 5; (**d**) N = 10.

**Figure 3 polymers-14-02382-f003:**
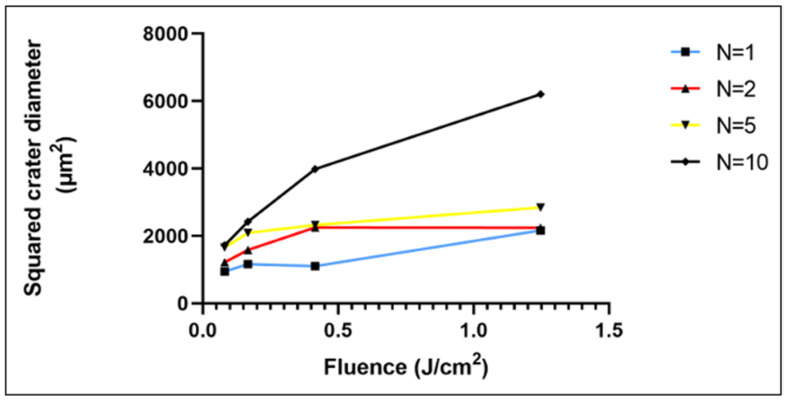
Increase in laser crater diameter with rising laser fluence and applied laser pulses.

**Figure 4 polymers-14-02382-f004:**
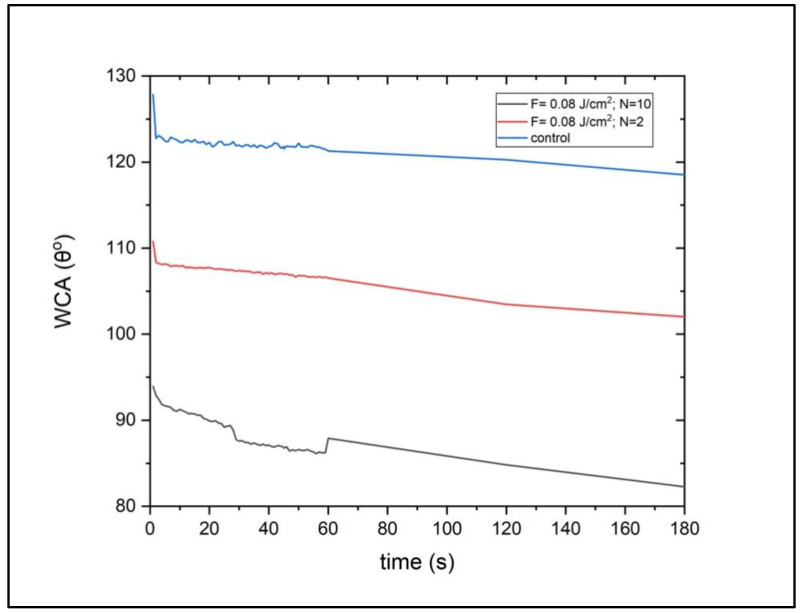
Water contact angle evaluation before and after fs-laser treatment. The scaffolds processed with F = 0.08 J/cm^2^ and N = 10 showed an improved wettability behaviour (θ = 94° at t = 1 s; θ = 82.3° at t = 180 s) when compared to the control ones (θ = 123.3° at t = 1 s; θ = 115.8° at t = 180 s).

**Figure 5 polymers-14-02382-f005:**
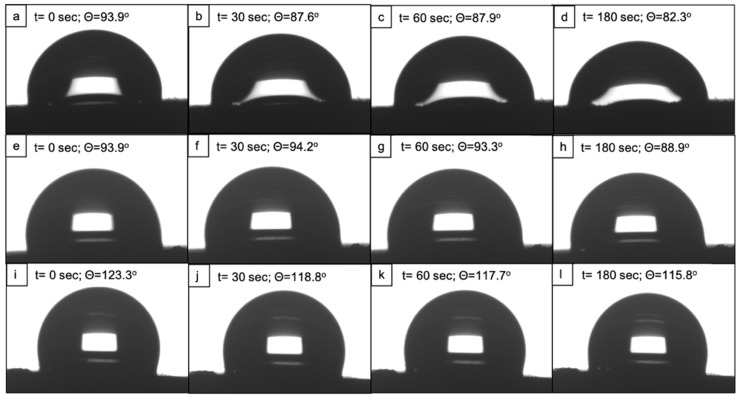
Evolution of a water droplet on the surface of fs-laser processed and control scaffolds for a period of 3 min. First-row images (**a**–**d**) represent a water droplet on scaffolds irradiated with F = 0.08 J/cm^2^; N = 10; second-row images (**e**–**h**) represent a water droplet on scaffolds treated with F = 0.08 J/cm^2^; N = 2; third-row images (**i**–**l**) depict the hydrophobic nature of the control PCL scaffold.

**Figure 6 polymers-14-02382-f006:**
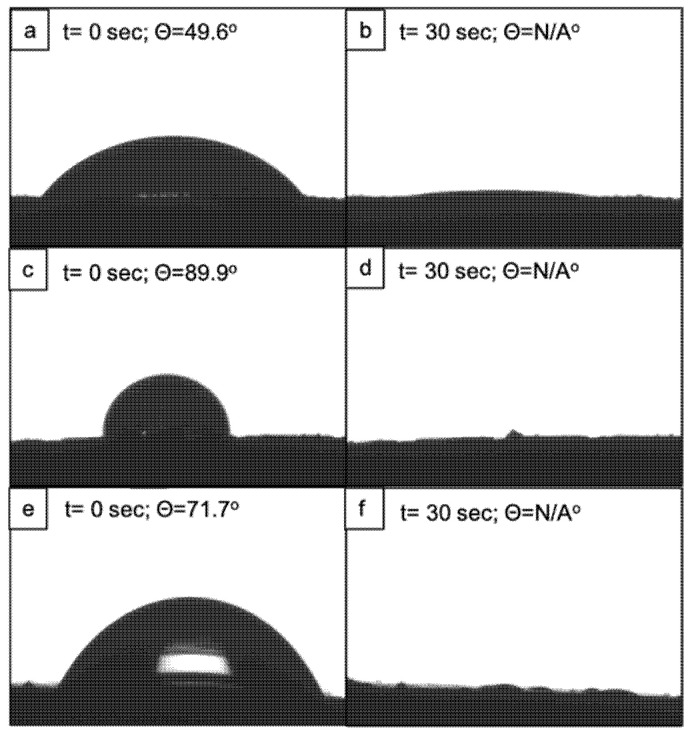
Contact angle of ethylene glycol on fs-laser-treated PCL. (**a**,**b**) Surface with microchannels (F = 0.08 J/cm^2^; N = 10); (**c**,**d**) Surface with microprotrusions (F = 0.08 J/cm^2^; N = 2); (**e**,**f**) Control scaffold.

**Figure 7 polymers-14-02382-f007:**
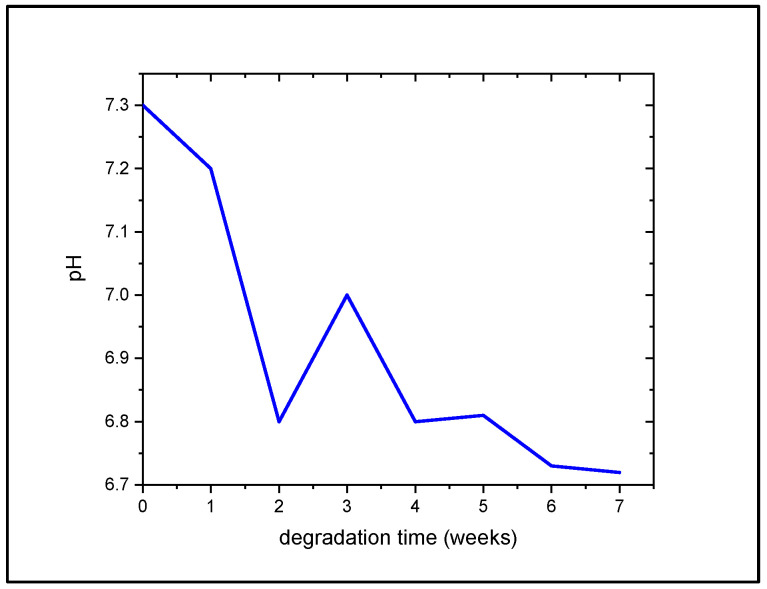
Monitoring pH change as a response of fs-laser-treated PCL degradation in PBS over 7 weeks.

**Figure 8 polymers-14-02382-f008:**
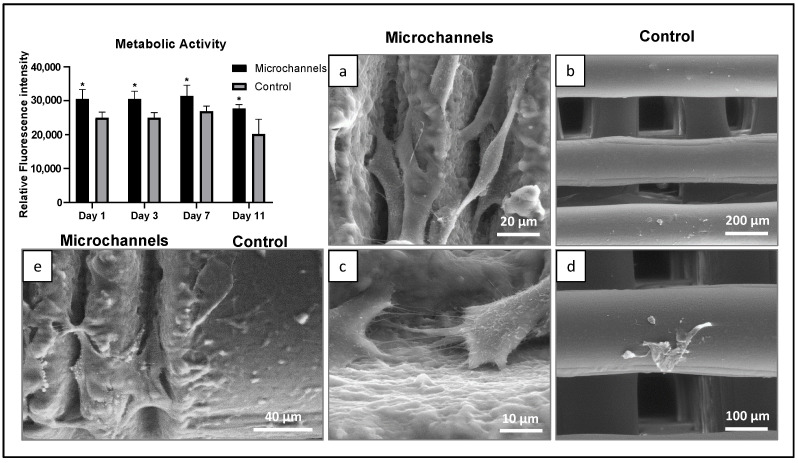
Metabolic activity (days 1, 3, 7, and 11) and SEM representative images (day 7) of human osteoblastic cells cultured over laser-induced microchannel topography on PCL (**a**,**c**,**e**) and control (**b**,**d**,**e**), in basal conditions. Scale bar: 20 μm (**a**); 200 μm (**b**); 10 μm (**c**); 100 μm (**d**), and 40 μm (**e**). * Significant difference from control (unmodified PCL), *p* ≤ 0.05.

**Figure 9 polymers-14-02382-f009:**
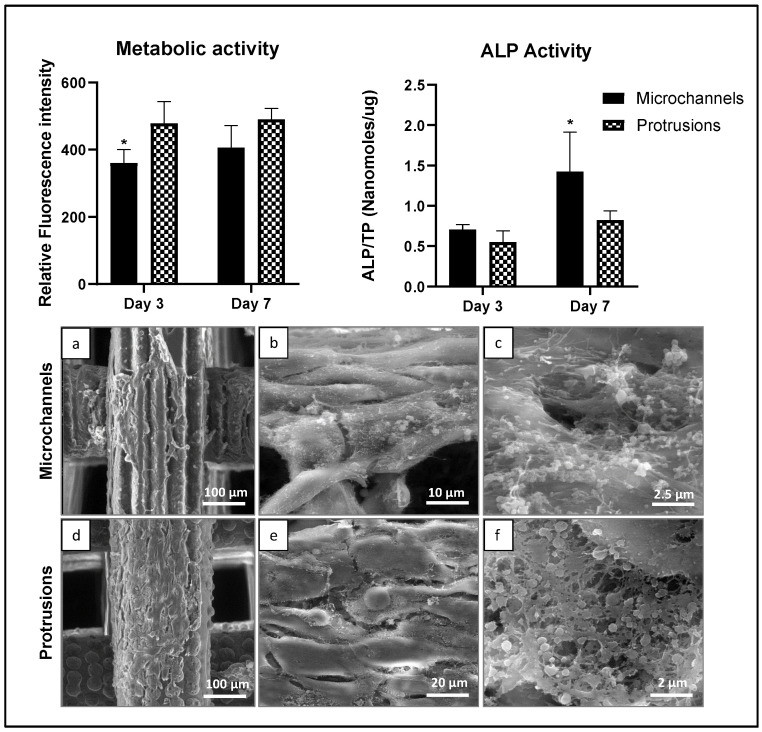
Metabolic activity (days 3 and 7), ALP activity (days 3 and 7), and SEM representative images (day 7) of human osteoblastic cells cultured over laser-induced topographies on PCL microchannels (**a**–**c**) and protrusions (**d**–**f**), in osteogenic conditions. Scale bar: 100 μm (**a**,**d**), 10 μm (**b**), 20 μm (**e**), 2.5 μm (**c**), and 2 μm (**f**). * Significant difference from protrusions, *p* ≤ 0.05.

**Figure 10 polymers-14-02382-f010:**
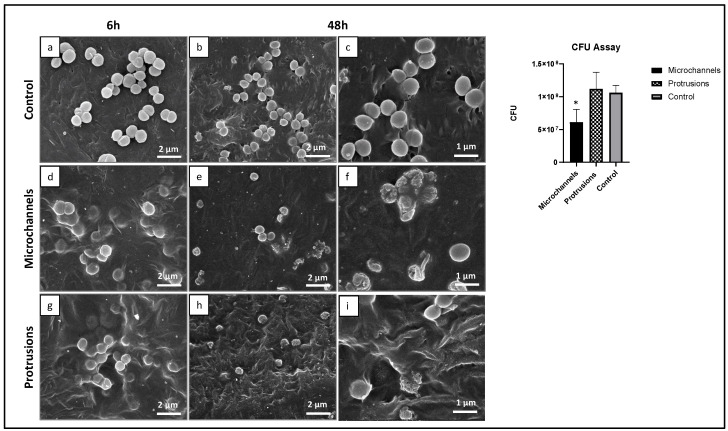
Representative SEM images (6 h and 48 h) of PCL surfaces colonized with *S. Aureus*—topographies of the control (**a**–**c**), laser-induced microchannels (**d**–**f**), and protrusions (**g**–**i**),. CFU assay on the three PCL surfaces. Scale bar: 2 μm (**a**,**b**,**d**,**e**,**g**,**h**), and 1 μm (**c**,**f**,**i**). * Significant difference from control (unmodified PCL), *p* ≤ 0.05.

**Figure 11 polymers-14-02382-f011:**
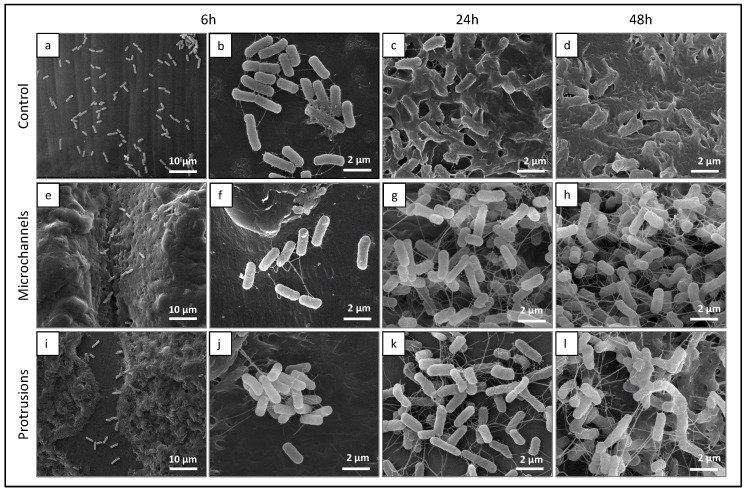
Representative SEM images (6 h, 24 h, and 48 h) of PCL surfaces colonized with *E. coli*—topographies of the control (**a**–**d**), laser-induced microchannels (**e**–**h**), and protrusions (**i**–**l**). Scale bar: 10 μm (**a**,**e**,**i**), and 2 μm (**b**–**d**,**f**–**h**,**j**–**l**).

## Data Availability

Not applicable.

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
