# Peer review of "Investigating Potential Effects of Ultra-Short Laser-Textured Porous Poly-ε-Caprolactone Scaffolds on Bacterial Adhesion and Bone Cell Metabolism"

_polymers, 2022, doi:10.3390/polym14122382_

Round 1

Reviewer 1 Report

The submitted manuscript is well written, and the reported work provides novel insights into the topic. The authors have presented a good work for consideration, e.g., the results and discussion section is impressive, and the findings are supported with references.

Reviewer 2 Report

In this study, the authors have treated 3D printed PCL scaffolds with ultra-short laser irradiation for surface patterning and analyzed its influence on attachment and proliferation of osteoblastic cells as well as S. aureus and E. coli cells. The results of the study showed that the micro-channels improved the attachment, growth, and proliferation of osteoblastic cells. However, the same topography had an inhibitory effect more on S. aureus than E. coli. Although the study presents interesting results and written in a nice manner, there are couple of points that need to be addressed before the publication of the manuscript.

1. On Page 9 and in lines 303-307, the authors report a substantial decrease in surface free energy (from 194 mN/m to 20 mN/m), which would mean that the surfaces have become more hydrophobic (which is not the case). Also, the authors mention that the laser treatment reduced the hydrophobic CH2 groups, which means that the surface energy should go up not go down. My guess is that since the laser treatment induces roughness, the OWRK method does not provide correct estimates of surface free energy. Also, the measured contact angles are almost 0 for ethylene glycol and diidomethane, which means one cannot use these results for surface energy calculation.

2. The authors should discuss why the micro channels promote osteoblastic cell growth and inhibit S. aureus growth.

Reviewer 3 Report

Authors need to write bacterial species names in italics.

Line 239: What do you mean by Ra? Authors need to write the full form in the sentence “A Ra measurement……...”

Line 274 to 276: “As we have shown before, the fs laser treatment reduced the hydrophobic CH2 groups which could explain the tendency for slightly lower contact angle”. The author needs to provide a proper mechanism in their article on how the laser treatment reduced the hydrophobic CH2 group rather than citing the reference.

Line 292 to 295: “Contrary to the results with distilled water, ethylene glycol drops fully spread over the surface of both the laser-treated and the control samples approximately 30 seconds after deposition (Figure 6). In the case of the highly dispersive solution (diiodomethane), the drops of the solvent immediately wetted the whole surface of both modified and control scaffolds.” Authors must provide the reasons why this distinct behavior for ethylene glycol and diiodomethane was noticed.

Line 302 to 303: “it was noted that the fs laser irradiation reduced greatly the surface free energy of the material”. Provide a reason for the statement.

Line 344: “S. aureus adhesion was observed on all PCL surfaces (microchannels, protrusions, and non-laser treated control), but colonization appeared relatively limited.” Need proper justification for this observation.

Line 354: “CFU analysis further revealed that the microchannels showed potential antibacterial activity compared to protrusions.” Why do microchannels reveal better antibacterial activity than protrusions?

The samples were bacteriostatic or bactericidal? The authors need to mention both the samples (microchannels and protrusions).

Line 361: “Behavior of E. coli on PCL samples was clearly different. SEM analysis revealed a significantly higher bacterial coverage compared to that of S. aureus.” Mention the proper reason for the statement.

In addition to the wettability test, antibacterial test, cellular adhesion, and proliferation the author also needs to perform the mechanical testing of the laser-treated samples in the revised manuscript. As mechanical stability is considered a key property for bone tissue engineering.

The surface property of the sample changes from time to time as it comes directly in contact with the external environment. The authors need to provide information about the surface modification consistency (time duration).

During bone tissue regeneration the degradation of the scaffold occurs. Thus, degradation affects the surface property of the sample. The authors must perform the degradation test in a physiological environment (simulated body fluid) and predict the changes in the surface morphology..

Round 2

Reviewer 3 Report

The authors have addressed all the comments and included all the required information in the revised manuscript.